# Image-to-Image Translation Using Identical-Pair Adversarial Networks

**Thai Leang Sung** [1] **and Hyo Jong Lee** [1,2,*]

1   Division of Computer Science and Engineering, Chonbuk National University, Jeonju 54896, Korea
2   Center for Advanced Image and Information Technology, Chonbuk National University, Jeonju 54896, Korea
*   Correspondence: hlee@chonbuk.ac.kr; Tel.: +8210-2680-5119

**Abstract:** We propose Identical-pair Adversarial Networks (iPANs) to solve image-to-image translation problems, such as aerial-to-map, edge-to-photo, de-raining, and night-to-daytime. Our iPANs rely mainly on the effectiveness of adversarial loss function and its network architectures. Our iPANs consist of two main networks, an image transformation network T and a discriminative network D. We use U-NET for the transformation network T and a perceptual similarity network, which has two streams of VGG16 that share the same weights for network D. Our proposed adversarial losses play a minimax game against each other based on a real identical-pair and a fake identical-pair distinguished by the discriminative network D; e.g. a discriminative network D considers two inputs as a real pair only when they are identical, otherwise a fake pair. Meanwhile, the transformation network T tries to persuade the discriminator network D that the fake pair is a real pair. We experimented on several problems of image-to-image translation and achieved results that are comparable to those of some existing approaches, such as pix2pix, and PAN.

**Keywords:** Image transformation; image-to-image translation; generative adversarial network; image synthesis

---

## 1. Introduction

Image-to-image translation refers to a constrained synthesis task of automatically transforming an input image to a synthetic image or mapping an input image to the desired output image. There are several applications of image-to-image translation in the fields of image processing, computer graphics, and computer vision, that transform an input image and generate high-resolution RGB images. e.g. image in-painting [1], image colorization [2], image super-resolution [3], image segmentation [4], and image denoising [5].

Recently, convolutional neural networks have been used for various image-to-image translation tasks [6–9]. The convolutional neural networks were trained to find the mapping from the input image to the transformed image by penalizing the discrepancy between the output and ground-truth images. Those approaches achieved their performance differently depending on cost function designs. Many efforts have been made to design effective losses and to improve the quality of performance by using pixel-wise losses, but they have received only blurry results [7,8]. Producing sharp and realistic images is an open challenge and requires better approaches.

To overcome this challenge, pix2pix-cGANs [10] trained models for image-to-image transformation by performing adversarial training to distinguish between real and fake images and by using generative adversarial losses to evaluate the discrepancy between a generated image and the real image. According to their results, they claimed that generative losses are more beneficial to the existing CNNs approaches. However, conditional adversarial networks (cGANs) are not stable, and their objective function must depend on a pixel-wise L1 loss function to make the generated output image closer to the ground-truth

image. Though the author claimed to use L1 instead of L2 to avoid blurry issues, their results are still blurry and limited to human vision in the realm of realistic. Perceptual Adversarial Networks (PAN) [11] used a feature-matching loss from the discriminator by adding feature distance to the generative adversarial loss. PAN produces better results but still has problems similar to those of pix2pix.

While the above issues are still existing an issue of unaligned paired image in a dataset has occurred. There is a limit of gathering a pair image for training its tasks such as rain removal and night-to-day. We are not always able to collect pair image of one event in the same location or environment. Unsupervised image-to-image translation (UNIT) [12] is an early approach for translating an image from one domain to another without any corresponding images in two domains. Dual-GAN [13] intended to develop a learning framework for unlabeled image data as unsupervised dual learning. [14] proposed disentangled representations to handle unpaired training data. This approach uses cross-cycle consistency loss. Domain adaptation technique was proposed by [15] to address the domain shift issue by discovering a mapping from the source data distribution to the target distribution. In a study by [16], Cycle-consistent Adversarial Networks (cycleGAN) [16] has a solution to the problem of unpaired-image translation as well. [16] generates input image forward and backward and uses cycle consistent function to verify the outputs.

In our approach, we propose iPANs to improve the quality of generated images and to reduce their blurriness. We want realistic and sharper results. The iPANs consist of two networks, an image transformation network T and a discriminative network D. We use U-NET [17] as network T and a perceptual similarity network [18] as the discriminator network D. Therefore, our discriminator requires two input images, for example, a generated image and a ground-truth image. The pix2pix-cGANs architecture [19] assumed the input image and ground-truth image to be a real pair. However, in our work, a real pair is made up of the duplicate input of the real sample image or ground-truth image. We perform adversarial training by distinguishing between real identical and fake identical-pairs of images and allow them to play a minimax game through discriminative network D. Although the discriminator network D defines the transformed input and the ground-truth image as fake, the image transformation network T attempts to make them the real inputs. In addition, we discover deep feature distance between the generated image and the ground truth through perceptual similarity network. We also extend our work to applications for unpaired image-to-image translation as well.

In summary, our paper has the following contributions:

- Identical-pair adversarial networks as a new conditional adversarial networks approach are proposed.
- A perceptual similarity network is used as the discriminative network D to distinguish between the real and fake identical-pairs.
- We propose deep feature distance as a perceptual loss to penalize the discrepancy between the two inputs of discriminative network D.
- We extend our work to unpair image-to-image translation by introducing our iPANs to cycleGAN's framework, respectively.

Our results on several tasks of image-to-image translation have been reasonably better than those of existing approaches such as pix2pix and PAN.

In Section 2, we will describe related work, and in Section 3 we will illustrate our proposed methods in detail. We will exhibit our experimental setup and validation results in Section 4. We explain the extended work to the unpair image-to-image translation in Section 5. Finally, we will discuss our results and conclusion.

## 2. Related Work

A variety of methods using feedforward CNNs have been proposed for image-to-image translation tasks [20]. Feedforward CNNs are trained and updated by backpropagation. The feedforward model

can be run forward very fast in test time. Image semantic segmentation methods generate dense scene labels from a single input image [21–23]. Image de-raining methods aim to remove rain or snow from the scene [6]. Image inpainting attempts to recover an image that has a missing part(s) [8,24]. Feedforward CNNs approaches have also been applied to image colorization [25].

Recently, GANs-based approaches have been developed for image-to-image translation as well. A Generative Adversarial Network [26] learns a generative model, which generates samples from real-world data. A generative network and a discriminative network of GANs compete with each other by playing a minimax game. Through the competition, GANs can generate realistic images from random noise input. Since then, They have been proposed continuously, such as InfoGANs [27], Energy-based GANs [28], WGANs [29], Progressive GANs [30], and SN-GANs [31]. Their approaches used for image-to-image translation tasks include ID-cGANs [32] for image de-raining, iGANs [33] for an interactive application, the IAN [34] for photo modification, and Context Encoder for image inpainting [8]. The Pix2pix-cGANs were used to solve several image-to-image translation tasks, such as object edges image to picture, aerial pictures to maps, or semantic labels to street view. It was claimed to be the first generic image-to-image translation framework of GANs-based approaches [19]. PAN [11] is another approach that results in better quality of the generated images. PAN proposed perceptual loss in addition to the generative adversarial loss. Then cycleGANs [16] approached it one step further to both pair and unpair image-to-image translation.

Blurriness and realistic approximation at the human level are still challenges that require more study in this field.

## 3. Methods

We explain the fundamental concept of our proposed iPANs for image-to-image translation tasks. First, we will explain our framework's architecture and then the adversarial loss function we used for the architecture and deep feature loss function.

### 3.1. Proposed Network Architecture

Figure 1 is an example in which we applied our network architectures to a map→photo task. Our architecture consists of two networks, i.e., the image transformation network T, and the discriminative network D.

**Image transformation Network T.** There are two transformation networks have been recommended by [35], ResNet and U-NET. These two are very comparable and optional to use and task-dependable according to [35]. We just choose U-NET for our transformation network T and make it as a default standard of our work. We use it to generate the transformed image T(x) given the input image x. Like [11], our encoding network contains a stack of convolution, batchnorm, and LeakyReLu, and our decoding network contained deconvolution, batchnorm, and ReLu in each layer. Finally, we uses hyperbolic tangent (tanh) activation for the output of the transformed image without batchnorm. Table 1 shows the detailed parameters of network T.

**Discriminative Network D.** Our discriminative network D uses a perceptual similarity network [18] which were used to find perceptual metric in [18] by Richard Zhang. But in our approach, we uses this network for two benefits: (1) as a discriminator, and (2) as a perceptual network to find the distance between two inputs. The network contains two streams of networks that share the same weights and are concatenated at the last layer. A given y and transformed image T(x) are input to the network. The y' and T'(x) are the vectors results from the two shared-weights CNNs of the similarity network. The result of the input y and T(x) is a fake pair determined by the discriminator network D. In contrast, the result of inputting a duplicated y is a real pair determined by the same discriminator. Table 2 shows the detailed parameters of discriminative network D. Different from discriminators in GAN or cGANs, our network D discriminates between a fake identical-pair and a real identical-pair. D considers input T(x) and y as a fake identical-pair because only y and y are the

real pair. Meanwhile, T attempts to synthesize T(x) to be identical to the real sample y and fools the network D into thinking that T(x) and y are the real identical-pair.

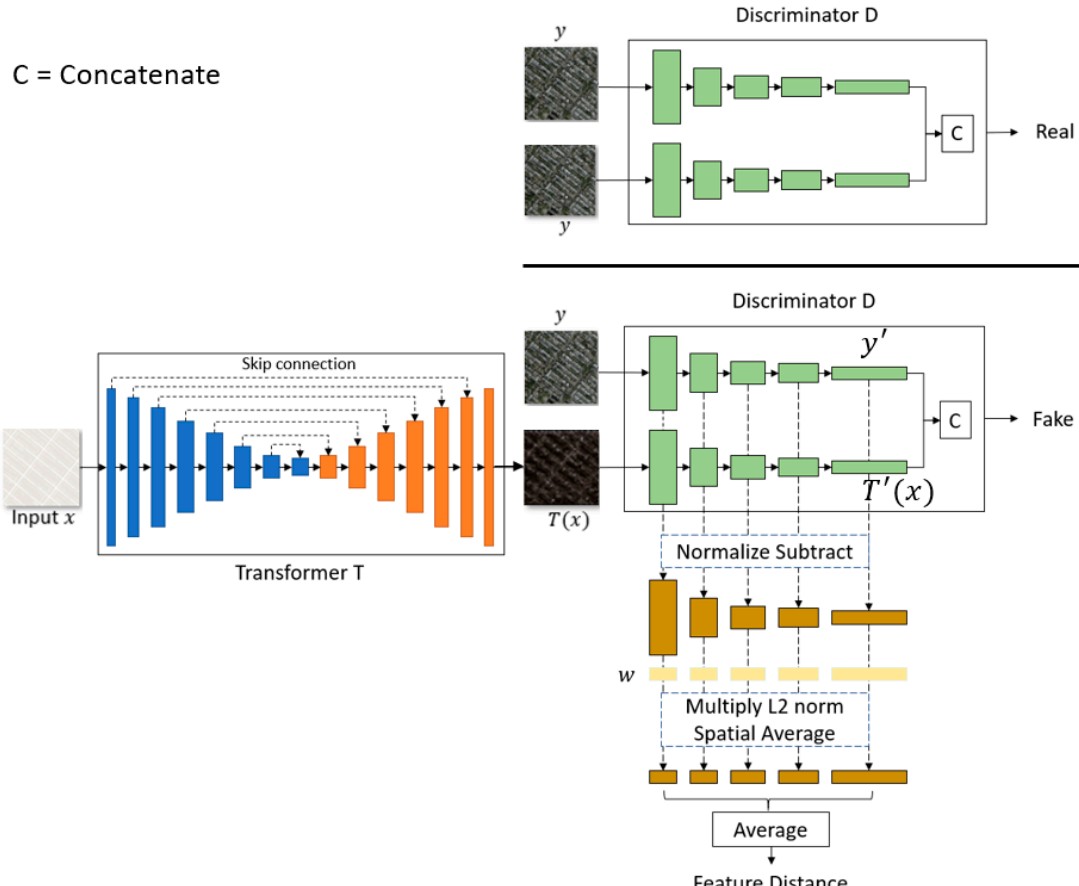

**Figure 1.** Proposed iPANs framework. It contains an image transformation network T and a discriminator network D. Given input x, network T synthesizes transformed image T(x) through U-NET. We use a perceptual similarity network in discriminator network D. We input ground-truth image y and transform T(x) to the discriminator, y→ y ′ and T(x)→T′(x), where y′ and T′(x) are to be concatenated. C means "concatenate". The Discriminator tends to distinguish a fake identical-pair from a real pair, i.e., y and T(x) ← fake, and y and y ← real. We find Feature Distance by subtracting each $y'_j$ of layer j with $T'_j(x)$ of layer j, normalizing, spatial averaging, and finally averaging all the scalar values from each layer. See the detail parameters of our network T and D in Tables 1 and 2.

**Perceptual Network.** As mentioned above, this perceptual network was originally used to find perceptual metrics in order to reach the level of human perception. Also, it used perceptual loss function in its approach finds the distance between features of two inputs from the chosen layer of the discriminator using L2. The discriminator network is the pre-trained model of VGG network that was trained using ImageNet which is still limited. In our work, we uses a pre-trained VGG trained by [18]. It was trained with several datasets that were manually evaluated by people in addition to ImageNet. Unlike perceptual loss in PAN, our perceptual loss doesn't calculate distance directly between each layer, but it computes deep embeddings of the two given networks, normalize the activations in the channel dimension, scales each channel by vector w, and take the L2 distance. We then average across spatial dimension and all layers. See Equation (5).

**Training Procedures.** To make our approach easily understood and followed. We show our practical training procedures as follows:

- Training the image transformation network T

a.  Input image x through T: x → T(x)
b.  Input T(x) and y through D: T(x), y → D(T(x), y)
c.  Find loss: L(T) (See Section 3.2)
d.  Find perceptual loss: Lp (see Section 3.2)
e.  Total loss: L(T) + Lp
f.  Backpropagation and update T

- Training discriminative network D

a.  Input T(x) and y through D: T(x), y → D(T(x), y)
b.  Find loss: $L_{fake}$ (See Section 3.2)
c.  Input duplicate y through D: y, y → D(y, y)
d.  Find loss $L_{real}$ (See Section 3.2)
e.  Total loss: $(L_{fake} + L_{real})/2$
f.  Backpropagation and update D

Our details on image-transformation network T and discriminative network D are listed in Tables 1 and 2.

**Table 1.** The architecture of the Image Transformation Network T.

| Image Transformation Network T | | | | | | |
|---|---|---|---|---|---|---|
| **Layers** | **Kernel** | **Stride** | **Channels** | **W × H** | **Activation** | **BN** |
| Input: Image | | | 3 | 256 × 256 | | |
| Conv. Layer 1 | 4 | 2 | 64 | 128 × 128 | LeakyReLU | |
| Conv. Layer 2 | 4 | 2 | 128 | 64 × 64 | LeakyReLU | True |
| Conv. Layer 3 | 4 | 2 | 256 | 32 × 32 | LeakyReLU | True |
| Conv. Layer 4 | 4 | 2 | 512 | 16 × 16 | LeakyReLU | True |
| Conv. Layer 5 | 4 | 2 | 512 | 8 × 8 | LeakyReLU | True |
| Conv. Layer 6 | 4 | 2 | 512 | 4 × 4 | LeakyReLU | True |
| Conv. Layer 7 | 4 | 2 | 512 | 2 × 2 | LeakyReLU | True |
| Conv. Layer 8 | 4 | 2 | 512 | 1 × 1 | LeakyReLU | |
| Deconv. Layer 9 | 4 | 2 | 1024 | 2 × 2 | ReLU | True |
| Concatenate (Layer 9, Layer 6) | | | | | | |
| Deconv. Layer 10 | 4 | 2 | 1024 | 4 × 4 | ReLU | True |
| Concatenate (Layer 10, Layer 5) | | | | | | |
| Deconv. Layer 11 | 4 | 2 | 1024 | 8 × 8 | ReLU | True |
| Concatenate (Layer 11, Layer 4) | | | | | | |
| Deconv. Layer 12 | 4 | 2 | 1024 | 16 × 16 | ReLU | True |
| Concatenate (Layer 12, Layer 3) | | | | | | |
| Deconv. Layer 13 | 4 | 2 | 512 | 32 × 32 | ReLU | True |
| Concatenate (Layer 13, Layer 2) | | | | | | |
| Deconv. Layer 14 | 4 | 2 | 256 | 64 × 64 | ReLU | True |
| Concatenate (Layer 14, Layer 1) | | | | | | |
| Deconv. Layer 15 | 4 | 2 | 128 | 128 × 128 | Tanh | |
| Output: transformed image | | | 3 | 256 × 256 | | |

**Table 2.** The architecture of the Discriminative Network D.

| Layers | Kernel | Stride | Channels | W × H | Activation | BN |
|---|---|---|---|---|---|---|
| **Discriminative Network D** | | | | | | |
| Input: image1, 2 to network streams | | | 3 | 256 × 256 | LeakyReLU | True |
| Conv. Layer 1 | 4 | 2 | 64 | 128, 128 | LeakyReLU | True |
| Conv. Layer 2 | 4 | 2 | 128 | 64, 64 | LeakyReLU | True |
| Conv. Layer 3 | 4 | 2 | 256 | 32, 32 | LeakyReLU | True |
| Conv. Layer 4 | 4 | 2 | 512 | 16, 16 | LeakyReLU | True |
| Conv. Layer 5 | 4 | 2 | 1024 | 8, 8 | LeakyReLU | True |
| Conv. Layer 6 | | | 1 | 8, 8 | | |
| Concatenate (stream1, stream2) | | | | | | |
| Output: real or fake pair | | | | | | |

Note: This network has two CNN streams, and they share the same parameters. Therefore, we show only one stream of information.

### 3.2. Proposed Loss Functions

$$\min_{G}\max_{D}V(G,\ D) = E_{y\sim p_{data}}[\log(D(y))] + E_{z\sim p_z}[\log(1 - D(G(z)))], \tag{1}$$

We will start with GANs here. A generative network G maps samples from input noise $p_z$ to real data $p_{data}$ through a training procedure of playing a minimax game with a discriminative network D. During training, the discriminator network D attempts to distinguish the real sample y~pdata from the generated image G(z). The generative network G aims to confuse network D by generating realistic images iteratively. Equation (1) is the formula of minimax game mentioned above, where then became generative adversarial loss:

In our methods, we adopt the GANs learning strategy to solve our tasks, because GANs-based methods recently have revealed their strong ability to learn generative models such as image generation [27,29,36]. As shown in Figure 1, we generated transformed image T(x) through network T gaven x ∈ X. Every input image x has a corresponding ground-truth image y, where y ∈ Y obeys the distribution $p_{data}$ real-world images. However, in our approach, network D considers two inputs as a real pair or fake pair. A real pair means a pair of identical images, e.g. y and y. In this logic, the discriminative network D considers the input images as a real identical-pair when both inputs are y and as a fake identical-pair when one is a generated image T(x) and the other is a real sample y. Therefore, based on GANs, we obtain our generative adversarial loss equation as below:

$$\min_{T}\max_{D}V(T,\ D) = E_{y\sim Y}[\log(D(y,\ y))] + E_{x\sim X, y\sim Y}[\log(1 - D(T(x), y))], \tag{2}$$

The transformation network T aims to fool the discriminator network D. Therefore, in practice, the transformation network needs to update itself by minimizing its loss function:

$$T^* = \operatorname{argmin}\left(E_{x\sim X, y\sim Y}[\log(1 - D(T(x), y))]\right), \tag{3}$$

$$T^* = \operatorname{argmax}\left(E_{x\sim X, y\sim Y}[\log(D(T(x), y))]\right), \tag{4}$$

where, T* is our objective function for transformation network T. We train T using Equation (4) rather than Equation (3), which always saturates too early [26].

Now our last loss function is a perceptual loss. Figure 1 of our proposed approach also shows how to obtain the distance between generated image $T(x)$ and the ground truth $y$ with discriminator network D which actually has two streams of VGG network. We extract feature stack from j layers and

unit-normalize in the channel dimension, which we designate as $y'_{jhw}$, $T'_{jhw}(x) \in \mathbb{R}^{H_j \times W_j \times C_j}$ for layer j. We scale the activation channel-wise by vector $w_j \in \mathbb{R}^{C_j}$ and compute the L2 distance.

$$L_p = \sum_j \frac{1}{H_j W_j} \sum_{h,w} \|w_j \odot (y'_{jhw} - T'_{jhw}(x))\|_2^2 \tag{5}$$

Finally, our objective function is:

$$T^* = \text{argmax}\left(E_{x \sim X, y \sim Y}[\log(D(T(x), y))]\right) + L_p \tag{6}$$

## 4. Experiments and Results

We evaluate the performance of our proposed iPANs on several image-to-image translation tasks, which are popular in the fields of image processing, computer vision, and computer graphics.

### 4.1. Experimental Setup

We used datasets that have been used in previous existing approaches such as pix2pix [35], BicycleGANs [37], ID-cGANs [32] and cycleGANs [16]; and we keep the size of their training and test sets the same their settings to make a fair comparison.

We worked on such datasets as:

- Dataset from pix2pix for Edges→images, Aerials→images, Labels→facades
- The dataset from BicycleGANs for Night→day
- The dataset from ID-cGANs for De-raining

Our experiments were trained on NVIDIA Titan X (Pascal) GPUs, and we used Python for our implementations. We set our optimizers such as Adam solver with learning rate 0.0002, bias1 0.5, and bias2 0.999. We set batch size and training epochs depending on the size of the dataset and tasks. We trained our tasks with an iteration of around 10 k and selected only the best results.

To illustrate the performance of our tasks, we conducted qualitative and quantitative experiments to evaluate the performance of the transformed image directly for the qualitative experiments. We used such metrics as Peak Signal-to-Noise Ratio (PSNR), Structural Similarity Index (SSIM) [38], Universal Quality Index (UQI) [39], and Visual Information Fidelity (VIF) [40] to evaluate the performance over test sets.

### 4.2. Experimental Results

Figure 2 displays our visual results of night-to-day, edge-to-photo, aerial-to-map, and de-raining using our model. We use a real raining image to test our model. Even though the raining image in the de-raining dataset was artificially created by the image editor, our approach produced a very nice clear de-raining on the real raining image, as shown in the figure.

We compared our results with some existing methods on several tasks depending on the best results for each approach, such as ID-cGANs, pix2pix, bicycleGANs, PAN, Dual-GAN, and cycleGAN.

In Figure 3, we compared our de-raining result with those of ID-cGANs and PAN. Our result looks sharper and clearer than did the results of ID-cGANs and PAN even after zooming in. The result of PAN de-raining looks smoother than ours, but it contains blurriness.

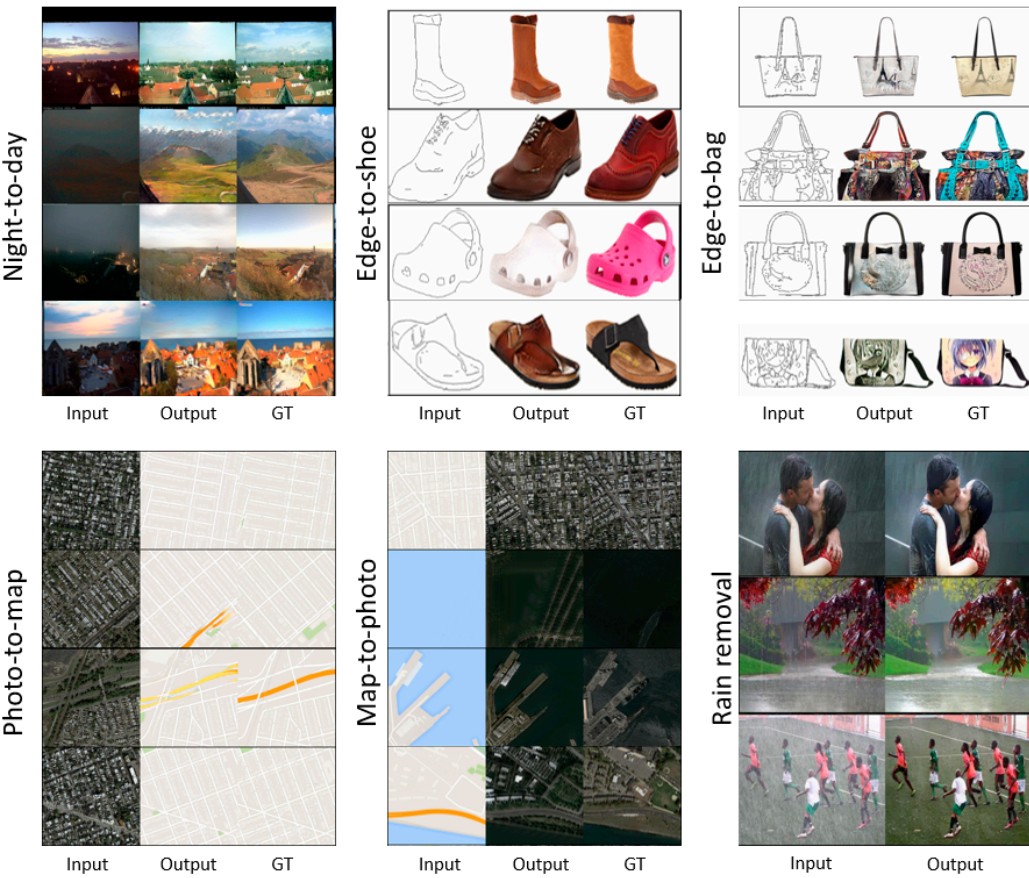

**Figure 2.** Sample results. We used our proposed iPANs to solve several tasks of image-to-image translation such as night-to-day, edge-to-photo, map-to-photo, photo-to-map, and de-raining.

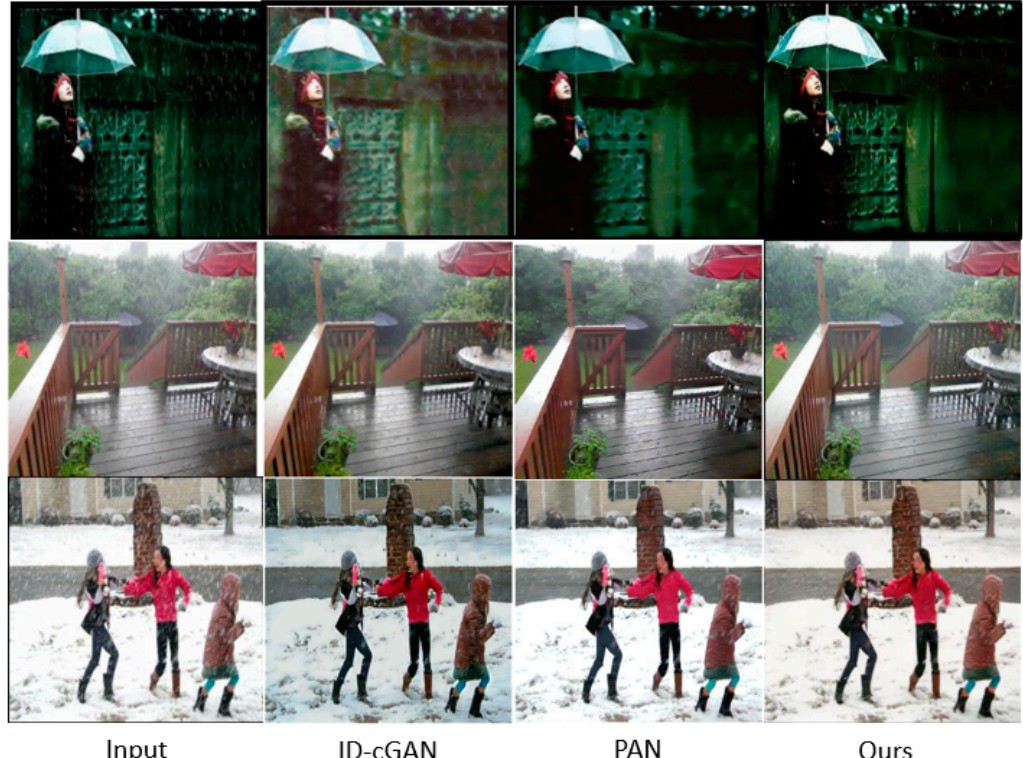

**Figure 3.** Comparison of de-raining tasks with ID-cGAN and PAN on a real rainy image.

Figure 4 compares the method with pix2pix, Dual-GANs, and PAN on the label→facade. The iPANs obtained a better visual facade than pix2pix but lost to PAN, because PAN produced a more realistic result. However, our results on aerial→map resembles the ground-truth image more than did ID-cGAN and PAN.

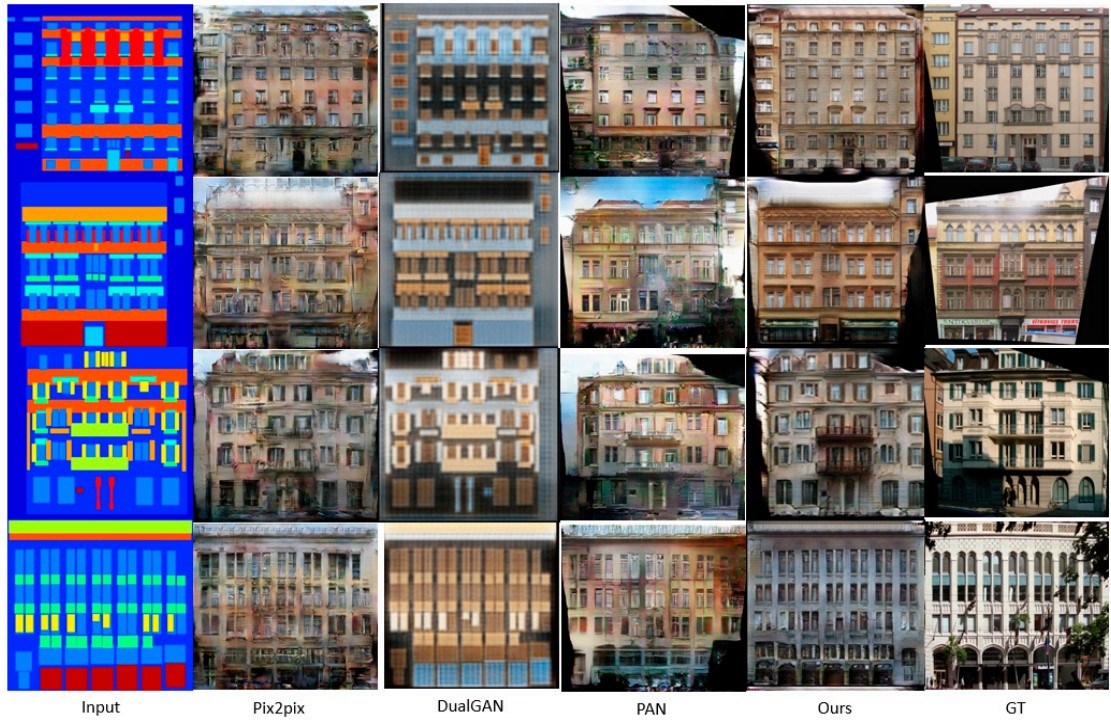

**Figure 4.** Comparison of the label→facade with Pix2pix, Dual-GAN, and PAN.

Visual-quality shows that nighttime to daytime results in Figure 5 look more realistic and brighter, like a daytime image, than did those of other methods. Pix2pix has a very blurry result, and still contains some darkness. BicycleGANs look realistic daytime but still remain a little bit dark.

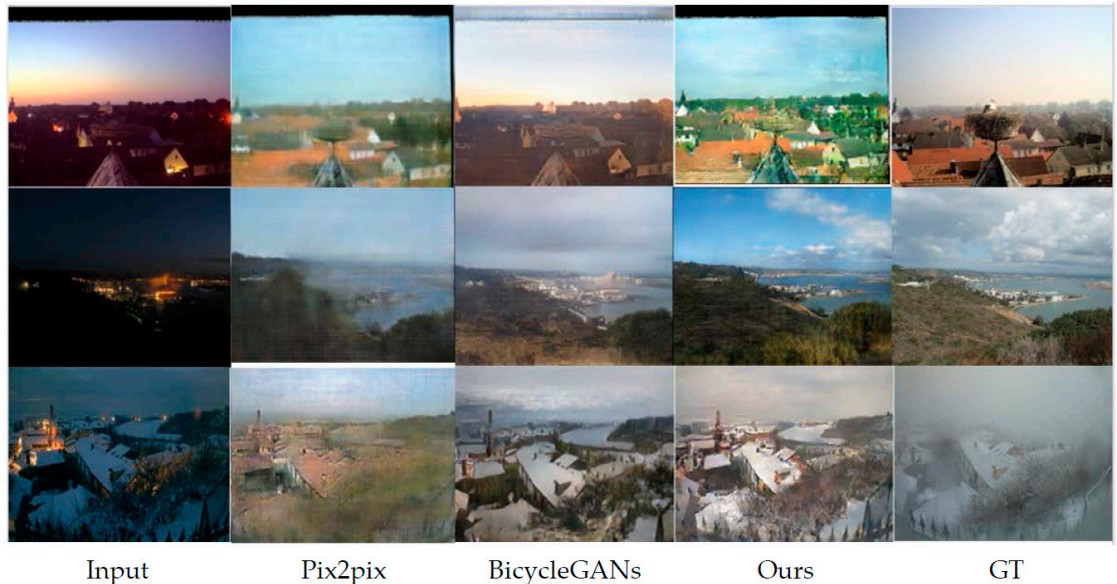

**Figure 5.** Comparison of night → day with pix2pix, BicycleGANs.

Finally, Figure 6 shows the results of satellite photo to map translation. Pix2pix and PAN produced curly road map while ours resembles the real sample images.

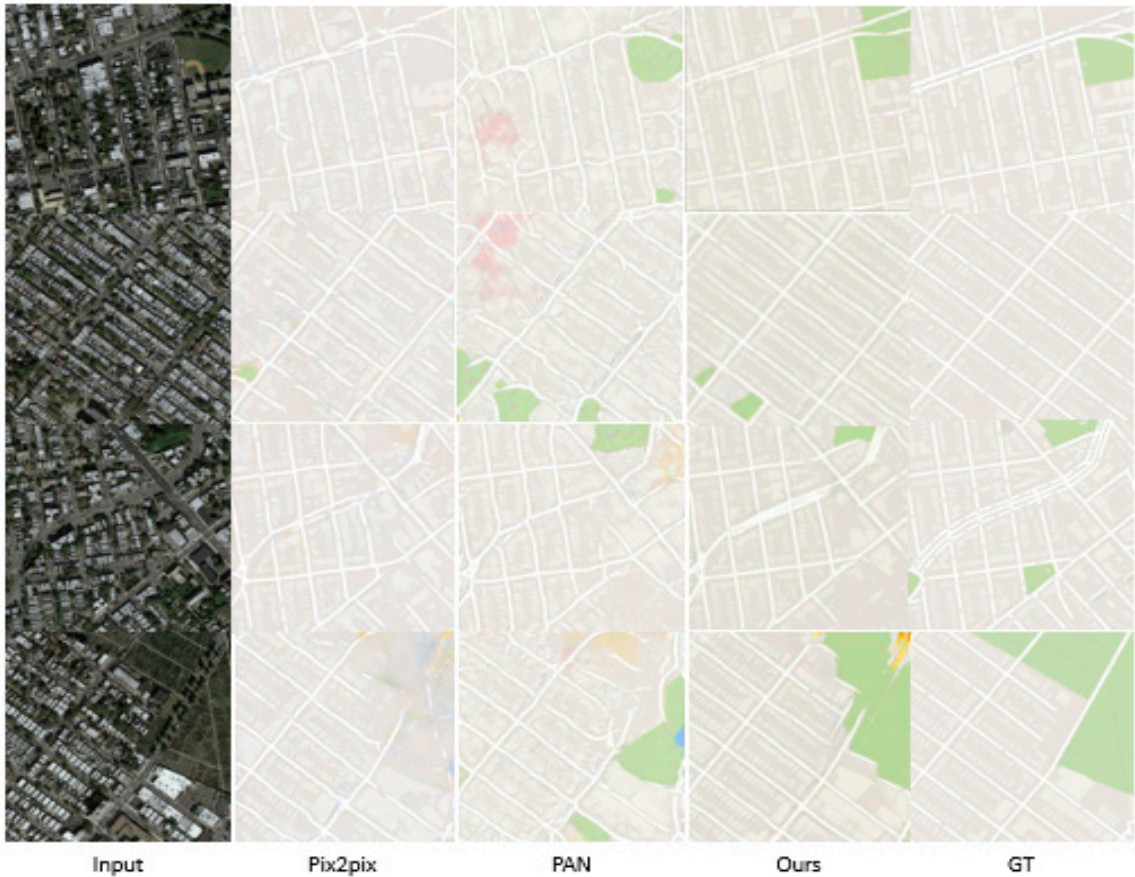

**Figure 6.** Comparison of the photo→map tasks with Pix2pix, PAN.

In Table 3, we exhibit our numerical results with existing methods through evaluation metrics. Our method on edge→shoes outperformed Pix2pix and PAN at PSNR 19.51 and SSIM 0.88 and UQI 0.94, and VIF 0.24. Our de-raining results achieved higher scores at PSNR 31.59, SSIM 0.89, UQI 0.78, and VIF 0.51 than did ID-cGANs or PAN. Our method on aerial→map outperformed pix2pix and PAN at PSNR 31.9, SSIM 0.79, UQI 0.64 and VIF 0.23. We obtained a higher score than pix2pix and PAN and Dual-GAN on the label→facade at PSNR 14.08 and UQI 0.86 and VIF 0.14, and lost to Dual-GAN at SSIM 0.53. Our task on the night→day obtained a higher score than pix2pix and bicycleGANs at PSNR 18.13 and UQI 0.88. Also, bicycleGANs outperformed ours at SSIM 0.64, and Pix2pix obtained the same result as ours at VIF 0.14.

As shown in Figure 1, we used U-NET as our image-transformation network and perceptual similarity network as our discriminator. To address whether our chosen networks are more beneficial, we used ResNet and basic CNNs for discriminator, then compared. In other words, we replaced image generative network (which is ResNet) and discriminator network used in Pix2pix [35] for our architecture's comparison. This comparison is to clarify why our choice of using U-NET (as an image-transformation network) and perceptual similarity network (as discriminator) is more suitable for our logical proposed approach.

**Table 3.** Comparison with Existing Work.

| Edges → Shoes | | | | |
|---|---|---|---|---|
| | **PSNR** | **SSIM** | **UQI** | **VIF** |
| Pix2pix | 15.74 | 0.42 | 0.07 | 0.05 |
| PAN | 19.51 | 0.78 | 0.34 | 0.23 |
| Ours | **21.01** | **0.88** | **0.94** | **0.24** |
| De-raining | | | | |
| | **PSNR** | **SSIM** | **UQI** | **VIF** |
| ID-cGAN | 22.91 | 0.81 | 0.64 | 0.38 |
| PAN | 23.35 | 0.83 | 0.66 | 0.40 |
| Ours | **31.59** | **0.89** | **0.78** | **0.51** |
| Aerial photos → Maps | | | | |
| | **PSNR** | **SSIM** | **UQI** | **VIF** |
| Pix2pix | 26.20 | 0.64 | 0.09 | 0.02 |
| PAN | 28.32 | 0.75 | 0.33 | 0.16 |
| Ours | **31.9** | **0.79** | **0.64** | **0.23** |
| Label→Facade | | | | |
| | **PSNR** | **SSIM** | **UQI** | **VIF** |
| Pix2Pix | 12.85 | 0.35 | 0.80 | **0.05** |
| PAN | 12.42 | 0.31 | 0.77 | **0.05** |
| Dual-GAN | 12.74 | **0.53** | 0.73 | **0.13** |
| Ours | **14.08** | 0.45 | **0.86** | 0.14 |
| Night → Day | | | | |
| | **PSNR** | **SSIM** | **UQI** | **VIF** |
| Pix2Pix | 9.46 | 0.57 | 0.76 | **0.14** |
| bicycleGANs | 17.33 | **0.64** | 0.60 | 0.12 |
| Ours | **18.13** | 0.54 | **0.88** | **0.14** |

According to the results shown in Figure 7, iPANs w/ G&DP was not balanced. This case happened to Pix2pix. We added L1 to our objective function (iPANs w/ G&DP + L1), but the results were blurry. The authors of pix2pix proposed additional L1 to their objective function and the results were also blurry. Then we replaced L1 with our perceptual loss (iPANs w/ G&DP + P). The results performed much better compared to the previous two models. However, iPANs w/ G&DP + P still cannot outperform our proposed architecture iPANs. According to the comparison results in Figure 7, our proposed model architecture is more beneficial while using U-NET and perceptual similarity network.

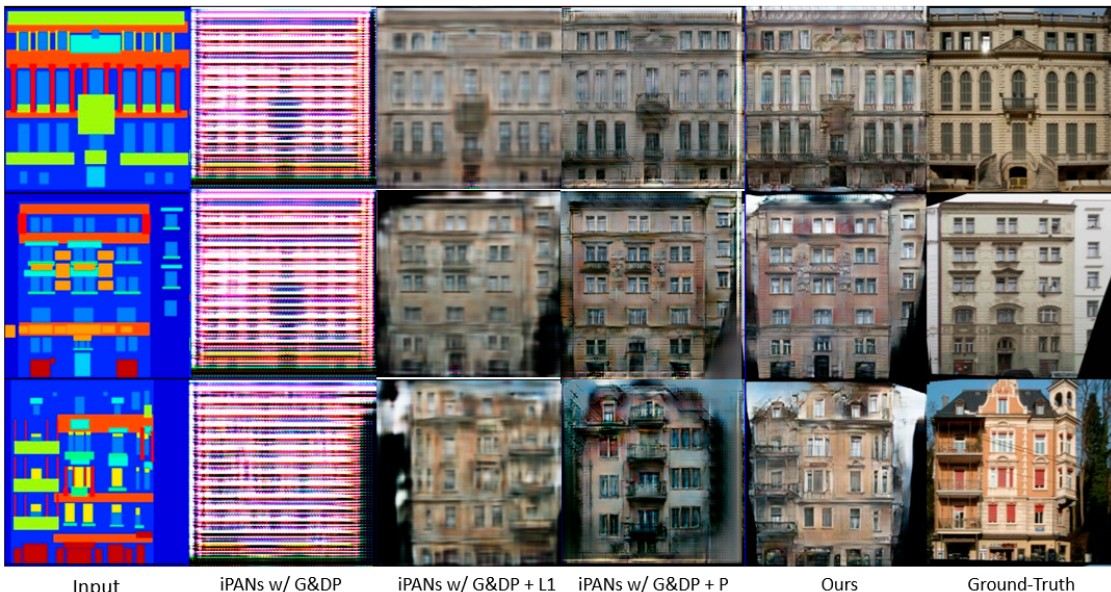

**Figure 7.** Comparison of iPANs with other networks.

## 5. Extension to Unpaired Image Translation

We mentioned in our related work section (Section 2) that there were some approaches have stepped a step further to solve some tasks that don't have pair image for training. Some works such as DualGan [13] and cycleGANs [16] investigated cross-domain image translations and performed the tasks without paired images.

We introduced our iPANs to cycleGAN in order to solve the issue of unpair image translation. This cycleGAN translates one domain to the other and translates back to where it started. The reversed version of the domain can arrive at where it started by using forward cycle-consistency loss and backward cycle-consistency loss. We can replace GANs used in cycleGAN with our iPANs because ours is a GANs-based approach as cycleGAN. So, we replace the GANs with our iPANs and obey the rule of cycleGAN's architecture, respectively. In this point, we excluded our perceptual loss from iPANs. We trained our extended work with horse ↔ zebra and exhibited some visual generated sample results in Figure 7. We keep all the code and the size of the training and test set of cycleGAN the same for a fair comparison, except adversarial loss which is related to the replacement of our iPANs.

## 6. Discussion

Our approach, as shown in Figure 1, explores a new network architecture similar to pix2pix but with a different logic. Our fake and real pairs are paired with ground-truth images, but cGANs are paired with the input image. A slight change of the architecture and adversarial loss method allows us to gain better results compared to those of cGANs-based pix2pix, ID-cGANs, Bicycle-GANs, and perceptual loss-based PAN methods. We extended our approach for unpair-image translation by replacing GANs in cycleGAN with our iPANs. However, during training, the networks didn't learn according to our purpose. After excluding perceptual loss function, the networks learned well, and we obtained the results as shown in Figure 8. Therefore, we will keep this issue for our future study. Instead of adapting with the concept of cycleGAN, our work should approach a better algorithm to solve the cross-domain translation.

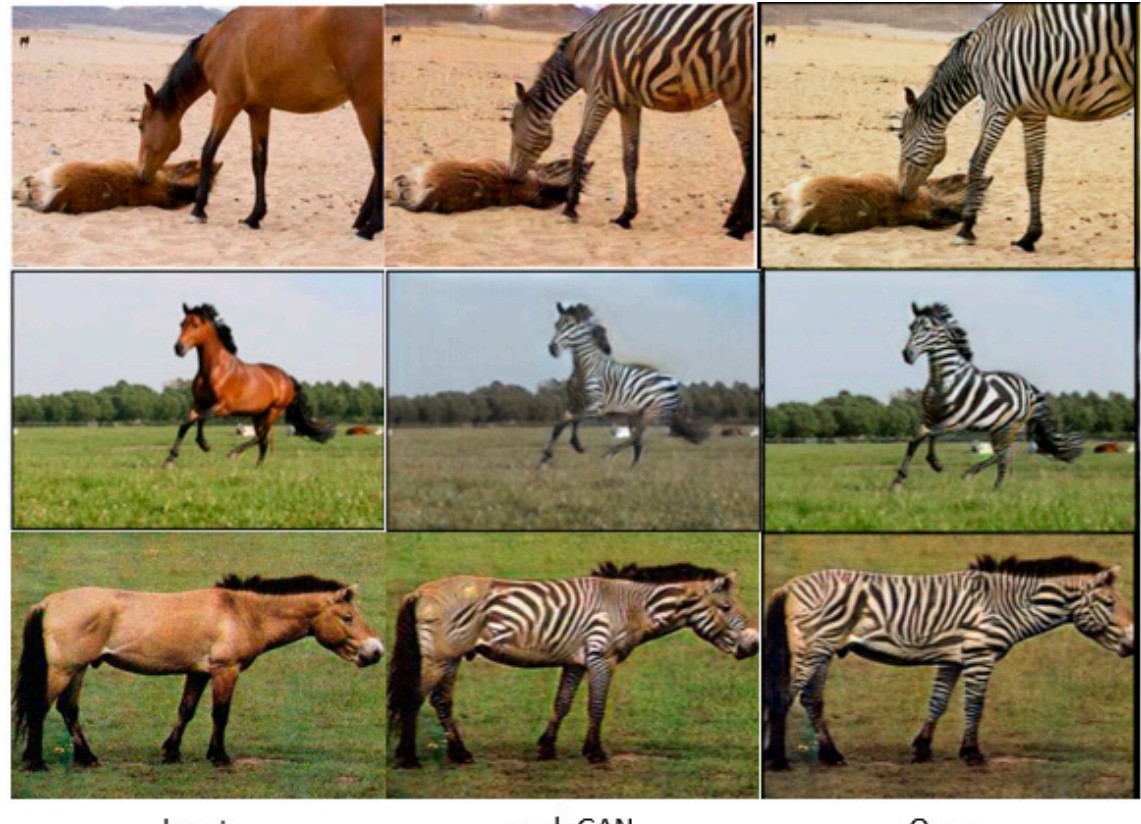

**Figure 8.** Sample results of the introduction of iPANs to a cycleGAN framework for unpair-image translation (horse ↔zebra).

## 7. Conclusions

We proposed iPANs that attempt to improve the image-to-image translation problems with new paired-input conditions for the replacement of conditional adversarial networks, i.e., identical images in which the ground-truth images are the real pair, but the generated image and ground-truth image are the fake pair. We introduce perceptual similarity network as our discriminator network and discover perceptual loss function at the same time through the network. We solved the unpair-image translation's problem by allowing cycleGAN to adapt our iPANs instead of its original GANs. We achieved very convincing results compared to those of existing methods.

**Author Contributions:** Conception and design of the proposed method: H.J.L and T.L.S.; Performance of the experiments: T.LS..; Writing of the paper:T.L.S.; Paper review and editing: H.J.L.

**Funding:** This research was supported by the Basic Science Research Program through the National Research Foundation of Korea (NRF) funded by the Ministry of Education (GR 2016R1D1A3B03931911).

**Conflicts of Interest:** The authors declare no conflict of interest.

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
