# Peer review of "Image-to-Image Translation Using Identical-Pair Adversarial Networks"

_applsci, doi:10.3390/app9132668_

Round 1

Reviewer 1 Report

This work proposes an algorithm for image to image translation. The idea is interesting, but there are several concerns needs to be addressed:

A-     The idea of transformed based im2im translation has been studied previously, and ideally this work should provide more technical contributions.

B-     It seems that this work requires a pair of images in two domain to train the model shown in Figure1. This would impose some limitation in the amount of training data collection, specially for some of the applications shown in the experimental results, such as rain removal, or day to night. Ideally we want to have some algorithms such as CycleGan which do not need paired images from both domain. Can the authors extend this framework to an unpaired im2im framework?

C-     The experimental results is not very well presented. First of all, there should be more explanation of the training data used to train this model, such as the size of training set, validation, and test set.

D-     Also, there should be a more careful discussion on the impact of different model parameters and hyper-parameters on the final results. I barely see such a discussion.

E-     There is not enough comparison with many of the recent state-of-the-art works on this problem. This paper fails to compare this work with promising works such as CycleGAN, DualGAN, etc. They need to provide a detailed comparison with all of these works in Table 3.

F-      Also, more example images are needed for visual comparison between various works, I would suggest to provide the visual comparison between this work and previous works, at least on 4 images for each task (in a separate figure).

G-     This paper is too short for a journal paper, it is more suited for a conference paper in the current format, unless the authors extends the method and experimental result sections.

H-     How did the authors decide to use U-Net for transformation network? Did they try different models and this one turns out to be the best? There should be a discussion on this in experimental results.

I-       Many of the recent works on im2im translation, and image generation based on GAN are missing from the introduction and references. I would suggest to do a more comprehensive literature study and add more relevant works to the references. Some of the relevant works on above topics are suggested below:

[1] "Dualgan: Unsupervised dual learning for image-to-image translation." Proceedings of the IEEE international conference on computer vision. 2017.

[2] "Diverse image-to-image translation via disentangled representations." Proceedings of the European Conference on Computer Vision (ECCV). 2018.

[3] "Image to image translation for domain adaptation." Proceedings of the IEEE Conference on Computer Vision and Pattern Recognition. 2018.

[4] "Iris-GAN: Learning to Generate Realistic Iris Images Using Convolutional GAN." arXiv preprint arXiv:1812.04822, (2018).

[5] Liu, Ming-Yu, Thomas Breuel, and Jan Kautz. "Unsupervised image-to-image translation networks." Advances in Neural Information Processing Systems. 2017.

[6] "Efficient Super Resolution for Large-Scale Images Using Attentional GAN." International Conference on Big Data (Big Data). IEEE, 2018.

[7] "Photo-realistic single image super-resolution using a generative adversarial network." Proceedings of the IEEE conference on computer vision and pattern recognition. 2017.

Author Response

Dear Reviewer,

We submit our revised version of the manuscript entitled " Image-to-Image Translation Using Identical Pair Adversarial Networks" (Manuscript ID: applsci-496941) for consideration to be published in Applied Sciences.

We thank you and the reviewers for their valuable comments. We have done our best to fully address their suggestions in the revised manuscript. All the modifications are marked with red color font..

A-   The idea of transformed based im2im translation has been studied previously, and ideally this work should provide more technical contributions.

Response: Our study provides contributions as mentioned in our Section 1 as the following:

·           Identical-pair adversarial networks as a new conditional adversarial networks approach is proposed.

·           A perceptual similarity network is used as the discriminator network D to distinguish between the real and fake identical pairs.

·           We propose deep feature distance as a perceptual loss to penalize the discrepancy between the two inputs of discriminator network D.

·           We extend our work to unpair image-to-image translation by introducing or iPANs to cycleGAN’s framework respectively.

B-     It seems that this work requires a pair of images in two domain to train the model shown in Figure1. This would impose some limitation in the amount of training data collection, specially for some of the applications shown in the experimental results, such as rain removal, or day to night. Ideally we want to have some algorithms such as CycleGan which do not need paired images from both domain. Can the authors extend this framework to an unpaired im2im framework?

Response: We have extended the framework to an unpaired im2im framework in a whole new Section 5 using the concept of cycle-consistency like cycleGAN. In practise, we just replace GAN in cycleGANs with our iPANs.

C-     The experimental results is not very well presented. First of all, there should be more explanation of the training data used to train this model, such as the size of training set, validation, and test set.

Response: We have added some description on how we set the training sets and the test sets according to the references that we used at line 213-215.

D-     Also, there should be a more careful discussion on the impact of different model parameters and hyper-parameters on the final results. I barely see such a discussion.

Response: Our methods don’t use extra parameters or hyper-parameters. We exhibit our network parameters in Table 1 and 2.

E-     There is not enough comparison with many of the recent state-of-the-art works on this problem. This paper fails to compare this work with promising works such as CycleGAN, DualGAN, etc. They need to provide a detailed comparison with all of these works in Table 3.

Response: We have added DualGAN to Figure 4 for Label2facade comparison, and its quality result in Table 3. We trained unpair-image translation extension of horse2zebra and visually compare the result with cycleGAN in figure 7.

F-      Also, more example images are needed for visual comparison between various works, I would suggest to provide the visual comparison between this work and previous works, at least on 4 images for each task (in a separate figure).

Response: We have updated the visual comparison between various works such in Figure 3, Figure 4, Figure 5, and Figure 6.

G-     This paper is too short for a journal paper, it is more suited for a conference paper in the current format, unless the authors extends the method and experimental result sections.

Response: We have extended our architecture, method and upgraded our experimental results. You can see our extended architecture in Figure 1 and our additional loss function [1] in method Section 3.1 and 3.2. All the sample results and comparison were retrained and improved.

Reference: [1] Richard ZhangPhillip IsolaAlexei A. EfrosEli ShechtmanOliver Wang; “The Unreasonable Effectiveness of Deep Features as a Perceptual Metric”. In CVPR, 2018.

H-     How did the authors decide to use U-Net for transformation network? Did they try different models and this one turns out to be the best? There should be a discussion on this in experimental results.

Response: We follow pix2pix on this issue. Pix2pix used two type of generators, ResNet and U-NET. Since these two networks are very comparable in im2im translation and depend on the tasks that they train, either one of them are just optional.

I-       Many of the recent works on im2im translation, and image generation based on GAN are missing from the introduction and references. I would suggest doing a more comprehensive literature study and add more relevant works to the references.

Response: We have added more relevant references related to unpair-image translation in the introduction on page 2 line 50 to 56. We also described how dataset are limited to pair image translation as below.

While the above issues are still existing, an issue of unaligned pair image in dataset has occurred. There is a limitation of gathering pair image for training its tasks like rain removal and night-to-day. We are not always able to collect pair image of one event in the same location or environment. Daul-GAN [12] intended to develop a learning framework for unlabeled image data as an unsupervised dual learning. Cycle-consistent Adversarial Networks (cycleGAN) [13] has a solution to this problem. [13] generates input image forward and backward and uses cycle consistent function to verify the outputs.

Reviewer 2 Report

The paper presents an image to image translation CNN using identical pair of adversarial Networks.

The papers provides information on background research but it will be beneficial to the reader that the related work is presented with a more critical perspective. For example there is a criticism of the L1 and L2 cost function used, but there is no commentary of the use of perceptual loss function. 

The  paper concludes that in the future they will try more reliable loss functions. However, no insight is given as for the reason the siamese (identical) discriminator provides better results compared to other work reviewed.

On page 5 the authors state that they selected only their best results. I am not clear what this means, and whether the experimental results presented includes a summary of the network's overall performance.  I believe that more information is needed on the testing of the architectures, the authors need to provide more information on the datasets used and how the results were chosen for comparison.

Author Response

Dear Reviewer,

We submit our revised version of the manuscript entitled " Image-to-Image Translation Using Identical Pair Adversarial Networks" (Manuscript ID: applsci-496941) for consideration to be published in Applied Sciences.

We thank you and the reviewers for their valuable comments. We have done our best to fully address their suggestions in the revised manuscript. All the modifications are marked with red color font.

Review 2:

The paper presents an image to image translation CNN using identical pair of adversarial Networks.

A -The papers provide information on background research but it will be beneficial to the reader that the related work is presented with a more critical perspective. For example there is a criticism of the L1 and L2 cost function used, but there is no commentary of the use of perceptual loss function. 

Response: We don’t use pixel-wise cost function in our approach. Furthermore, in our revised version, we extend our approach by adding a perceptual similarity network where its roles are discriminator and the creator of perceptual cost function.

B- The paper concludes that in the future they will try more reliable loss functions. However, no insight is given as for the reason the siamese (identical) discriminator provides better results compared to other work reviewed.

Response: We have extended this similarity network to something higher which is call perceptual similarity network. It was first proposed to find similarity for evaluation metric by Richard Chang.

C- On page 5 the authors state that they selected only their best results. I am not clear what this means, and whether the experimental results presented includes a summary of the network's overall performance.  I believe that more information is needed on the testing of the architectures, the authors need to provide more information on the datasets used and how the results were chosen for comparison.

Response: At this point I referred it to the references of the previous works that have been done on the same topic of image2image translation. How they choose the result and how the training and test sets were set. I described the refences at line 213-215

Round 2

Reviewer 1 Report

Thanks to the authors for submitting the revised paper and addressing some of my concerns.

There are still some concerns which have not been addressed.

The authors need to provide more comparison on the impact of different parameters, model architecture on the final results. They did not give a clear answer in their response.

Also some of the relevant works are missing.

Author Response

Dear reviewer

We submit our revised version of the manuscript entitled "Image-to-Image Translation Using Identical Pair Adversarial Networks" (Manuscript ID: applsci-496941) for consideration to be published in Applied Sciences.

We thank you for your valuable comments. We have done our best to fully address their suggestions in the revised manuscript. All the modifications are marked with red color font.

The following are the replies to the comments suggested by the reviewers.

Reviewer #1

Thanks to the authors for submitting the revised paper and addressing some of my concerns.

There are still some concerns which have not been addressed.

The authors need to provide more comparison on the impact of different parameters, model architecture on the final results. They did not give a clear answer in their response.

Also, some of the relevant works are missing.

Response:

Thank you for your valuable comment.

Our model architecture is basically using u-net and perceptual similarity network. Therefore, the number of parameters and their values are varied according to the basic networks that we use for our model. We compare the use of our proposed image transformation network and discriminator network with the generative network (Resnet9) and discriminator network which is the basic CNNs start with the concatenation of two inputs. We can see how different the results are when we use those two networks in our architecture instead of our chosen networks. Our proposed logic remains the same because we only want to know why our chosen U-net and similarity network are more suitable than others. The items that we compared:

1.      IPANs using Generative network and Discriminator of pix2pix (IPANs w/ G&DP)

2.      IPANs + Generative network and Discriminator used in pix2pix + pixel-wise loss (IPANS w/ G&DP+L1)

3.      IPANs + Ours with Generative network and Discriminator of pix2pix + perceptual loss (IPANs w/ G&DP+P)

4.      Our IPANs

We trained on facades dataset and set hyperparameters for this training process the same as we set in our experimental setup (Section 4.1). The results are as shown in Figure 7.

    As shown in Figure 1, we used U-NET as our image-transformation network and perceptual similarity network as our discriminator. To address whether our chosen networks are more beneficial, we used ResNet and basic CNNs for discriminator, then compared. In other words, we replaced image generative network (which is ResNet) and discriminator network used in pix2pix [35] for our architecture’s comparison. This comparison is to clarify why our choice of using U-NET (as an image-transformation network) and perceptual similarity network (as discriminator) is more suitable for our logical proposed approach.

    According to the results shown in Figure 7, iPANs w/ G&DP was not balanced. This case happened to pix2pix as well. We added L1 to our objective function (iPANs w/ G&DP + L1), but the results were blurry. The authors of pix2pix proposed additional L1 to their objective function and the results were also blurry.  Then we replaced L1 with our perceptual loss (iPANs w/ G&DP+P). The results performed much better compared to the previous two models. However, iPANs w/ G&DP+P still cannot outperform our proposed architecture iPANs. According to the comparison results in Figure 7, our proposed model architecture is more beneficial while using U-NET and perceptual similarity network.

Also, we have added more references to our related works section.

While the above issues are still existing, an issue of unaligned pair image in a dataset has occurred. There is a limitation of gathering pair image for training its tasks such as rain removal and night-to-day. We are not always able to collect pair image of one event in the same location or environment. Unsupervised image-to-image translation (UNIT) [12] is an early approach for translating an image from one domain to another without any corresponding images in two domains. Dual-GAN [13] intended to develop a learning framework for unlabeled image data as unsupervised dual learning. [14] proposed disentangled representations to handle unpaired training data. This approach uses cross-cycle consistency loss. Domain adaptation technique was proposed by [15] to address the domain shift issue by discovering a mapping from the source data distribution to the target distribution. Cycle-consistent Adversarial Networks (cycleGAN) [16] has a solution to the problem of unpaired-image translation. [16] generates input image forward and backward and uses cycle consistent function to verify the outputs.

References:

12.           Liu, M.-Y.; Breuel, T.; Kautz, J. Unsupervised image-to-image translation networks. In Proceedings of Advances in Neural Information Processing Systems; pp. 700-708.

13.           Yi, Z.; Zhang, H.; Tan, P.; Gong, M. Dualgan: Unsupervised dual learning for image-to-image translation. In Proceedings of Proceedings of the IEEE international conference on computer vision; pp. 2849-2857.

14.           Lee, H.-Y.; Tseng, H.-Y.; Huang, J.-B.; Singh, M.; Yang, M.-H. Diverse image-to-image translation via disentangled representations. In Proceedings of Proceedings of the European Conference on Computer Vision (ECCV); pp. 35-51.

15.           Murez, Z.; Kolouri, S.; Kriegman, D.; Ramamoorthi, R.; Kim, K. Image to image translation for domain adaptation. In Proceedings of Proceedings of the IEEE Conference on Computer Vision and Pattern Recognition; pp. 4500-4509.

16.           Zhu, J.-Y.; Park, T.; Isola, P.; Efros, A.A. Unpaired image-to-image translation using cycle-consistent adversarial networks. arXiv preprint arXiv:.03126 2017.

36.           Lotter, W.; Kreiman, G.; Cox, D. Unsupervised learning of visual structure using predictive generative networks. arXiv preprint arXiv:.06380 2015.
